# Spatial proteomics defines the content of trafficking vesicles captured by golgin tethers

John J. H. Shin [1✉], Oliver M. Crook [2,3,4], Alicia C. Borgeaud [1], Jérôme Cattin-Ortolá [1], Sew Y. Peak-Chew [1], Lisa M. Breckels [3], Alison K. Gillingham[1], Jessica Chadwick[1], Kathryn S. Lilley[2,3] & Sean Munro [1✉]

Intracellular traffic between compartments of the secretory and endocytic pathways is mediated by vesicle-based carriers. The proteomes of carriers destined for many organelles are ill-defined because the vesicular intermediates are transient, low-abundance and difficult to purify. Here, we combine vesicle relocalisation with organelle proteomics and Bayesian analysis to define the content of different endosome-derived vesicles destined for the trans-Golgi network (TGN). The golgin coiled-coil proteins golgin-97 and GCC88, shown previously to capture endosome-derived vesicles at the TGN, were individually relocalised to mitochondria and the content of the subsequently re-routed vesicles was determined by organelle proteomics. Our findings reveal 45 integral and 51 peripheral membrane proteins re-routed by golgin-97, evidence for a distinct class of vesicles shared by golgin-97 and GCC88, and various cargoes specific to individual golgins. These results illustrate a general strategy for analysing intracellular sub-proteomes by combining acute cellular re-wiring with high-resolution spatial proteomics.

[1] MRC Laboratory of Molecular Biology, Francis Crick Avenue, Cambridge CB2 0QH, UK. [2] The Milner Therapeutics Institute, University of Cambridge, Cambridge CB2 0AW, UK. [3] Cambridge Centre for Proteomics, Department of Biochemistry, University of Cambridge, Cambridge CB2 1QR, UK. [4] MRC Biostatistics Unit, School of Clinical Medicine, University of Cambridge, Cambridge CB2 0SR, UK. ✉email: shin.jaehee@gmail.com; sean@mrc-lmb.cam.ac.uk

Amajor goal of cell biology is to define the logic of vesicular traffic between organelles of the secretory and endocytic pathways. These trafficking pathways control organelle identity, abundance and function, as well as the precise complement of proteins on the cell surface. The content of vesicle carriers contains information of both the cargoes that define each trafficking route and the specificity cues that determine their destination. Hence, deconvolution of intracellular trafficking would be greatly aided by knowing the complete content of vesicles that connect the various cellular compartments. Many vesicles have a short lifetime, are found in low amounts and are difficult to purify. Thus, other than synaptic vesicles, which can be purified in large amounts[1], the proteomes of transport vesicles that connect different organelles are incompletely defined. Accurate membrane traffic depends on tethers at destination organelles that selectively capture incoming transport vesicles[2,3]. The golgins, a large and ancient family of ubiquitously expressed long coiled-coil proteins of the Golgi apparatus, function as tethers that contribute to the specificity of membrane traffic by selectively capturing transport vesicles prior to SNARE-mediated fusion[4–7]. The specific vesicles captured by the golgins are likely to represent a major proportion of the overall vesicles destined for the Golgi. However, the content of these vesicles remains poorly defined and so they are ideal targets for the development of strategies of vesicle purification.

Each Golgi compartment is decorated with a distinct set of golgins that are anchored to the membrane via their C termini, with golgin-97, golgin-245, GCC88 and GCC185 found at the *trans*-Golgi in mammalian cells[7–9]. Mutation of these *trans*-Golgi golgins results in only mild phenotypes, suggesting redundancies between them that make it challenging to investigate their functions[10]. This problem was overcome by an in vivo assay for golgin tethering that tested whether the golgins are sufficient, rather than necessary, to function as vesicle tethers[11]. Golgins were relocated to an ectopic site by swapping their C termini with a mitochondrial transmembrane domain (golgin-mito). This was found to be sufficient to redirect specific transport vesicles to the mitochondria and maintain them in a permanently tethered state (Fig. 1a). Using this assay, golgin-97-mito, golgin-245-mito and GCC88-mito specifically capture endosome-to-Golgi vesicles, with the mitochondrial TMD itself causing no detectable vesicle accumulation or mitochondrial stress[11,12]. Golgin-97 and golgin-245 share a conserved vesicle-binding motif at their N-termini which they use to capture the same pool of vesicles[13]. This motif binds directly to TBC1D23, a member of a family of Rab GTPase-activating proteins, and TBC1D23 bridges golgin-97 and golgin-245 to endosome-to-Golgi vesicles (Fig. 1a)[14]. TBC1D23 also binds to WDR11/FAM91A1/C17orf75, a protein complex of unknown function, although the precise role of this interaction remains unclear[14–16].

Many open questions regarding endosome-to-Golgi traffic remain. Multiple sorting nexins, as well as retromer and the clathrin adaptor AP-1 have been reported to act in this retrograde route but the number of distinct carriers and the precise role of the known trafficking components is uncertain[17–19]. Examining the retrograde transport vesicles could provide insight into these questions, but such vesicles are challenging to study as they are small and generally short-lived as they are rapidly consumed by fusion to the *trans*-Golgi. Ectopic capture of vesicles by golgins relocated to mitochondria provides a means to investigate their composition as the lack of relevant SNAREs on mitochondria means that the vesicles accumulate in a tethered state. Golgin-97 and golgin-245 have been reported to bind AP-1-derived vesicles through TBC1D23 and, possibly, the WDR11 complex; however, it is evident that AP-1 independent vesicles are also captured by these golgins[15]. In addition, the vesicle-associated specificity

factors that target transport vesicles to GCC88 are unknown, and the physiological roles that distinguish GCC88 from golgin-97 and golgin-245 are still poorly defined. To elucidate these questions, we set out to determine the content of the vesicles that are captured by golgin-97 and GCC88.

Applying in vivo proximity biotinylation to golgins relocated to mitochondria has been used successfully to identify direct binding partners and some vesicle cargo (Fig. 1a)[14]. However, its limited biotinylation range prevents this approach from determining the entire content of vesicles. Moreover, it is skewed against the identification of small proteins and proteins that have few exposed lysines to biotinylate, such as the short cytoplasmic tails of the transmembrane proteins that comprise the cargo of transport vesicles. In addition, mitochondrial purification by cell fractionation and immune-purification each come with their own caveats. Thus far, no method of cell fractionation achieves complete separation of organelles. Nor do they take into account the change in size and density of the mitochondria due to vesicle capture across treatments. In contrast, organelle proteomics methods such as the Localisation of Organelle Proteins by Isotope Tagging after Differential ultraCentrifugation (LOPIT-DC) method combines cell fractionation with quantitative mass spectrometry and multivariate data analysis to allow simultaneous characterisation of multiple sub-cellular compartments, without the requirement for total purification of compartments of interest (Fig. 1b, d)[20–24]. In this work, we opted to combine the mitochondrial relocation assay with LOPIT-DC proteomics to determine which vesicle cargo, adaptors and accessory proteins are redirected to mitochondria by golgin-97-mito and GCC88-mito. We find known cargo and many further transmembrane and peripheral membrane proteins to be associated with the vesicles captured by golgin-97, and find evidence for a distinct class of vesicle that can also be captured by GCC88. Together, we show that the combination of relocalisation of sub-cellular compartments and high-resolution spatial proteomics can be used as a general strategy to define intracellular sub-proteomes.

## Results and discussion
**LOPIT-DC on mitochondrial relocated golgin-97 and GCC88.** An overview of the LOPIT-DC work flow we used is shown in Fig. 1b, d. LOPIT-DC was applied to Flp-In 293 cells expressing golgin-97-mito, GCC88-mito or a mitochondrial TMD (mito) as a control (Fig. 1c). The profiles of approximately 6000 proteins present across three independent replicates for each treatment were obtained using multiplexing with conventional tandem mass spectrometry (MS²) (Supplementary Data 1). A prediction of the sub-cellular localisation for each protein was then made by matching individual profiles to profiles of known organelle markers using supervised machine learning with support vector machines (SVM), as well as using Bayesian statistical modelling through the T-Augmented Gaussian Mixture model (TAGM) method (Fig. 1d)[25]. The latter takes into account the uncertainty that arises when classifying proteins that reside in multiple locations, or unknown functional compartments and also those that dynamically move within the cell, and so provides a richer overall analysis of spatial proteomics data[25–28].

The performance of tandem mass tag (TMT)-based quantification by MS² can be affected by interference from contaminant peptides with similar properties to the target peptide, but if this does occur it has to be addressed by an additional round of ion selection and fragmentation in synchronous precursor selection mass spectrometry (SPS-MS³)[29]. Our LOPIT-DC data analysed by MS² showed effective separation of organelles with good resolution (Supplementary Fig. 1a, b), even though the overall

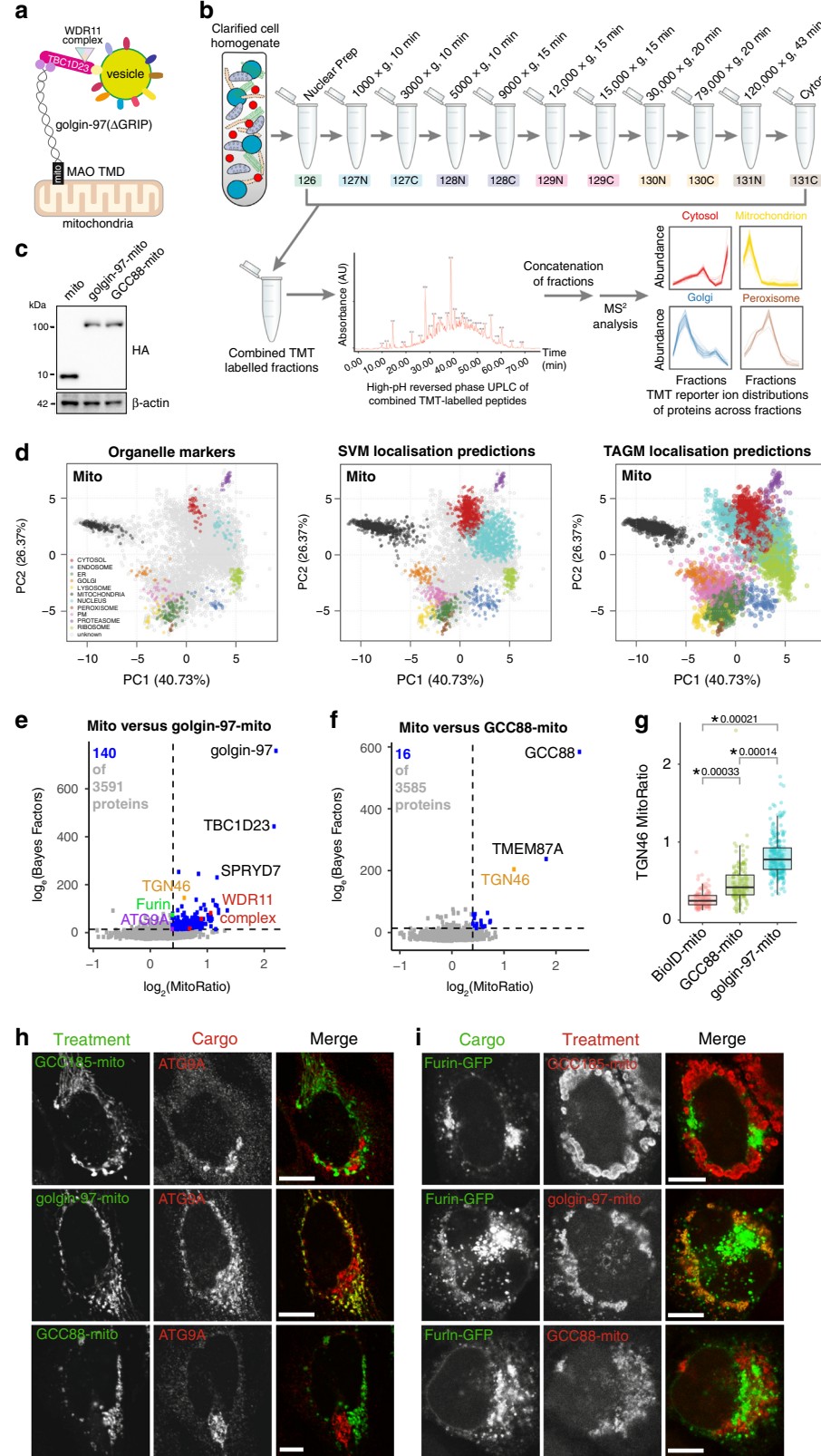

resolution of SPS-MS³ is somewhat better²³. Since our application of LOPIT-DC was not aimed at resolving subtle differences in the overall profiles of different organelles, but instead aimed at detecting clear changes in the profile of putative vesicle proteins between samples, our requirements were satisfied by using the simpler and less expensive MS² approach.

To identify proteins that translocate to mitochondria as a result of the expression of mitochondrial golgins, we initially leveraged the spatial information of our LOPIT-DC data by applying an independent pre-filtering step. Thus, proteins predicted by TAGM to localise to the mitochondria or nucleus in the mito control were discarded from subsequent analysis. These proteins

**Fig. 1 LOPIT-DC on mitochondrial relocated golgin-97 and GCC88. a** Applying proximity biotinylation to mitochondrially relocated golgin-97 identifies its interaction with TBC1D23 and the WDR11 complex on endosome-to-Golgi vesicles. **b** Overview of the LOPIT-DC work flow. Cells are gently lysed and fractionated by differential ultracentrifugation. The peptides of each fraction are then labelled with 11-plex TMT, pre-fractionated through high-pH reversed phase UPLC and then analysed by MS[2] or SPS-MS[3]. This gives the profiles of the proteome, which are then analysed as in **d**. Image is adapted from the LOPIT-DC methods paper[23], under a Creative Commons Attribution 4.0 International License [http://creativecommons.org/licenses/by/4.0/]. **c** Immunoblot of proteins from Flp-In 293 cells stably expressing the doxycycline-inducible mito, golgin-97-mito and GCC88-mito with 1 µg ml$^{-1}$ doxycycline for 48 h, representative of two independent experimental replicates. **d** Principle component analysis (PCA) projections for the LOPIT-DC showing organelle markers before classification, after SVM classification and after TAGM classification. **e, f** Results of MitoRatio versus Bayes Factor analysis comparing mito control to golgin-97-mito and GCC88-mito. Thresholds are based on experimentally confirmed hits (see text) ($\log_2$(MitoRatio) ≥ 0.40, $\log_e$(Bayes Factor) ≥ 14.0). Mito versus golgin-97-mito: 140 proteins in addition to golgin-97 were within threshold out of 5914 analysed proteins (3591 after pre-filtering), all 140 listed in Supplementary Data 2. Mito versus GCC88-mito: 16 proteins in addition to GCC88 were within threshold out of 5532 analysed proteins (3585 proteins after pre-filtering), all 16 listed in Supplementary Data 2. **g** Quantification of confocal micrographs of Flp-In 293 cells as used in LOPIT-DC (Supplementary Fig. 2a) measuring the ratio of the mean intensity of TGN46 at the mitochondria over its intensity at the Golgi (MitoRatio). Boxplots are of MitoRatios of all cells quantified across 3 independent experiments ($n ≥ 114$), and show the median, the first and third quartiles (Q1 and Q3), minimum (Q1-1.5 × interquartile range (IQR)), and maximum (Q3 + 1.5 × IQR). p-values are of two-sample two-sided $t$-tests comparing the overall mean MitoRatio of each replicate. Data are provided as a Source Data file. **h, i** Confocal micrographs of HeLa cells expressing the indicated golgin-mito construct (HA) stained for endogenous ATG9A or exogenous Furin-GFP, all micrographs representative of three independent experimental replicates. Scale bars, 10 µm.

are likely to be resident mitochondrial and nuclear proteins that are not relevant to our biological question. Discarding them made it easier to discern proteins that shift toward a mitochondrial profile in golgin-mito cell lines (Supplementary Fig. 1c).

We next analysed our data by quantifying general changes in protein profiles between golgin-mito cells and control cells using a Bayesian non-parametric two-sample test (Bayes Factor). To quantify protein profiles that shift towards a mitochondrial profile we computed a mitochondrial ratio (MitoRatio - see Methods). As expected, our strongest hits when comparing our LOPIT-DC of golgin-97-mito to the mito control were golgin-97 itself and TBC1D23 (Fig. 1e). Likewise, GCC88 was the strongest hit when comparing GCC88-mito to the mito control (Fig. 1f). Golgin-97-mito had more high scoring proteins than GCC88-mito, suggesting that golgin-97 has a greater overall ability to accumulate vesicles. This is consistent with our previous observations, and was further corroborated through quantitation of immunofluorescence microscopy (Fig. 1g and Supplementary Fig. 2a)[13].

We next set a threshold for protein relocation to the mitochondria based on confirmed experimental observations (Fig. 1e, f). The major criterion was that this threshold should have all subunits of the WDR11 complex (WDR11, FAM91A1 and C17orf75), as hits when comparing mito versus golgin-97-mito, but not when comparing mito versus GCC88-mito (Fig. 1a, e, f)[14]. Application of this threshold also predicted ATG9A and Furin, normally resident in Golgi membranes and endosomes, as cargo of vesicles captured by golgin-97-mito (Fig. 1e)[30,31]. We verified this by immunofluorescence (Fig. 1h, i), demonstrating that the vesicles that recycle them to the Golgi can be captured by golgin-97 but not GCC88 or GCC185. Together, these data demonstrate the effectiveness of our Bayesian analysis of the LOPIT-DC data.

**Cargo specific to golgin-97**. Proteins trafficking along the retrograde route to the Golgi are in a constant state of flux as they are continuously recycled throughout the endosomal network[17,32]. Consistent with this phenomenon, we observed a strong enrichment for proteins of the endosomal network amongst those affected by golgin-97-mito. This includes proteins classified by multiple localisation databases and by our LOPIT-DC analysis as being localised to the endosome, lysosome, Golgi or plasma membrane (PM), or present on vesicles

(Supplementary Data 2). Of these, a total of 45 transmembrane proteins were robustly affected, including IGF2R (CI-MPR), M6PR (CD-MPR) and TGN46, which are known cargo of vesicles captured by golgin-97 and golgin-245 (Fig. 2a, b)[13]. In addition, we also detected 5 SNAREs involved in the fusion of endosome-to-Golgi vesicles to the *trans*-Golgi. The majority of these transmembrane proteins had profiles similar to endosomes in our control, and then shifted to a more mitochondrial profile as a result of golgin-97-mito in our LOPIT-DC analysis (Fig. 2c, d). These shifts are partial, which is consistent with vesicles accumulating over the time course of the experiment with some cargo proteins still in transit, or in vesicles that have escaped capture by the mitochondrial golgins. Based on these results, we define these transmembrane proteins as cargo of vesicles captured by golgin-97.

Retrograde trafficking to the *trans*-Golgi is typically challenging to interrogate because of the range of machinery and pathways that have been proposed to be responsible for recycling specific cargo in specific vesicles[17,19,33]. Of these different pathways, the content of AP-1 derived vesicles is the best resolved[34–36]. We detected 16 protein that are known AP-1 cargo as being clearly affected by golgin-97-mito, consistent with previous reports of AP-1 vesicles binding to golgin-97 and golgin-245[15]. Of these, TVP23 is a poorly characterised TGN protein that has been implicated in endosome-to-Golgi trafficking in yeast, and secretion from the TGN in plants[37–39]. We found that TVP23B is specifically relocated to the mitochondria by golgin-97-mito and not GCC88-mito (Fig. 2e, f and Supplementary Fig. 2b). Furin is also a cargo of AP-1 that showed similar results (Fig. 1i). Together, this supports a conclusion that GCC88 has minimal ability to capture clathrin/AP-1-derived vesicles.

A significant complicating factor in the study of retrograde trafficking is that many cargo proteins contain multiple sorting motifs that allow adaptors from numerous sorting pathways to bind to them. This redundancy has made it challenging to conclusively determine the necessity of some of these pathways. In addition to AP-1 cargo, we also identified 29 other cargo that are likely to be in transport vesicles derived from other sorting pathways that are captured by golgin-97 and, by inference, golgin-245 (Fig. 2b). These proteins are thus candidates that may be exploited in the future to further elucidate these pathways. An example is the autophagy protein ATG9A and its partner SERINC1, which are known to be sorted by AP-4 adaptors into vesicles secreted from the TGN to the periphery

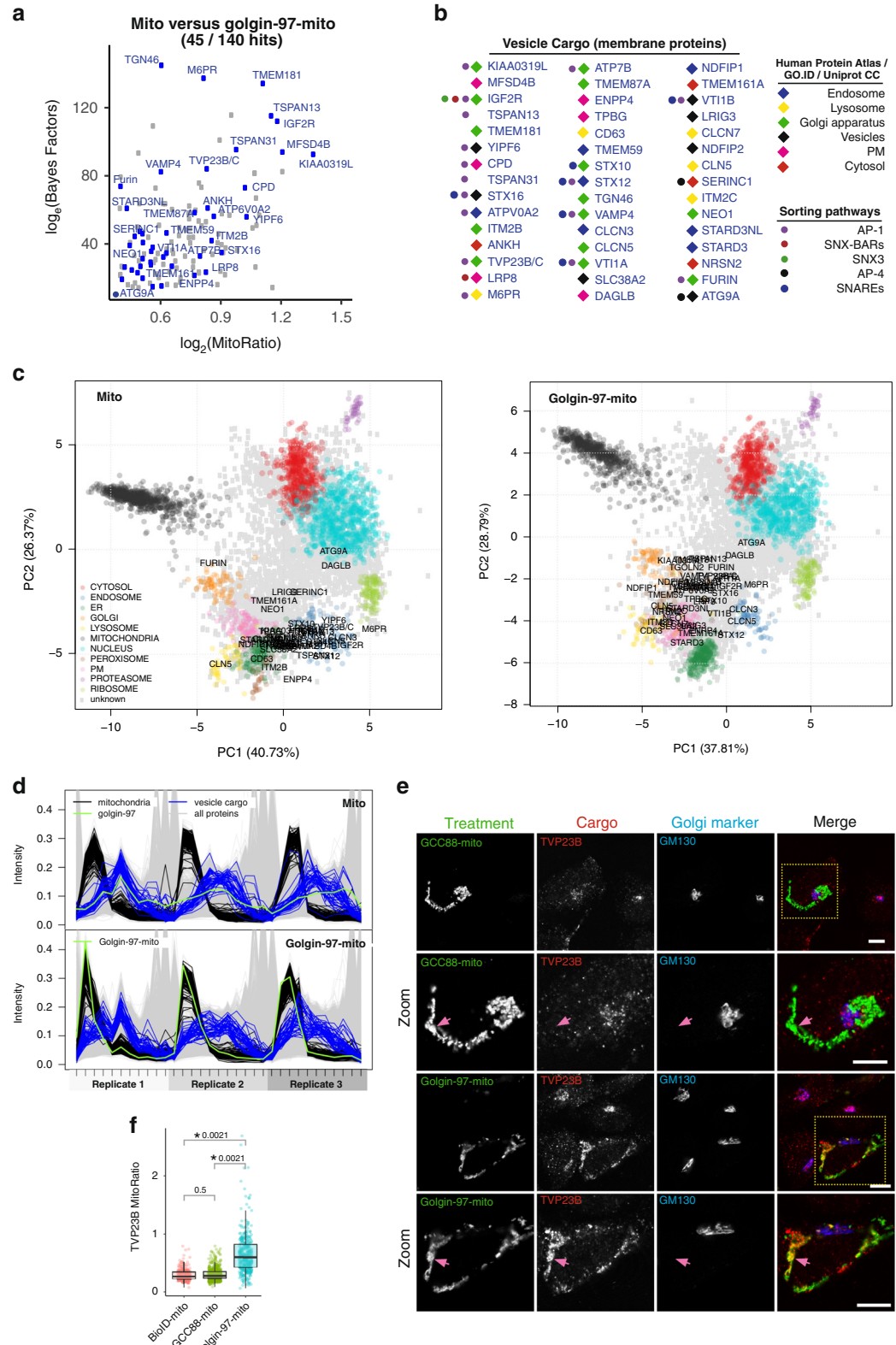

of the cell[40]. Importantly, our analysis did not detect AP-4 or its accessory proteins, RUSC1 and RUSC2. This strongly suggests that ATG9A and SERINC1 are on retrograde vesicles derived from an as yet undefined sorting pathway. ATG9A is an important regulator of autophagosome formation and SERINC proteins have recently been identified as HIV restriction factors[41,42]. Their relocation to mitochondria by golgin-97-

mito could provide a valuable assay for future studies on the role of these proteins.

**Adaptor proteins and accessory factors of vesicles specific to golgin-97.** The discovery of TBC1D23 and the WDR11 complex in Golgi tethering was enabled by proximity biotinylation of

**Fig. 2 Golgi/endosome membrane proteins of vesicles specific to golgin-97. a** MitoRatio versus Bayes Factor analysis of mito versus golgin-97-mito showing all hits (blue) that are both localised to the Golgi/endosomal network and are also transmembrane proteins. Axes start at the cut offs shown in Fig. 1e, ie $\log_2$(MitoRatio) = 0.40, $\log_e$(Bayes Factor) =14.0). **b** All hits shown in **a** ordered from left column to right column based on descending MitoRatio scores. Classification of their localisation and sorting pathway is based on multiple localisation databases and previous publications. **c** PCA projections for the LOPIT-DC of mito versus golgin-97-mito showing SVM classifications and all hits from **a**, **b**. **d** TMT reporter ion distributions of proteins across fractions for each replicate for mito versus golgin-97-mito showing the profiles of known mitochondrial markers (Mitochondria) and all hits from **a**, **b** (Vesicle Cargo). **e** Confocal micrographs of HeLa cells expressing the indicated golgin-mito construct (HA) stained for endogenous TVP23B (cargo) and GM130 (marker of cis-Golgi). All micrographs representative of three independent experimental replicates, scale bars 10 μm. **f** Quantification of micrographs of Flp-In 293 cells used in LOPIT-DC (Supplementary Fig. 2b) showing the ratio of TVP23B intensity at the mitochondria to intensity at the Golgi (MitoRatio). Boxplots show MitoRatios from three independent replicates, each of $n \geq 209$ cells, as for Fig. 1g. P-values are of two-sample two-sided t-tests comparing the overall mean of each replicate. Data are provided as a Source Data file.

vesicles captured by golgin-97-mito[14]. This suggests that peripheral membrane proteins relocated to the mitochondria by golgin-97-mito could be adaptor proteins and accessory factors involved in the tethering process. The localisations of 51 peripheral membrane proteins of the endosomal network were affected by golgin-97-mito (Fig. 3a, b), including VPS45, an SM (Sec1p / Munc18) protein involved in the fusion of endosome-to-Golgi vesicles with the TGN. We also found an enrichment for adaptor proteins and accessory factors from multiple endosomal sorting pathways, including AP-1 adaptors (AP1G1, AP1S2, AP1AR, AP1M1 and AP1S1) and accessory proteins (CLINT1, SYNRG, PI4K2B and Rab9), but not their associated clathrin coat[34,36,43,44]. We also found proteins involved in retromer-dependent transport (SNX3, SNX4, VPS35, VPS26, VPS29), along with SNX-BAR proteins potentially involved in retromer-independent transport (SNX1, SNX2, SNX5 and SNX6)[45,46]. Many peripheral membrane proteins are localised to multiple compartments in the cell, making them difficult to classify by spatial proteomics (Fig. 3c). However, we clearly observed that golgin-97-mito induced a movement of these proteins to a more mitochondrial profile (Fig. 3c-f). As with TBC1D23, we would expect that the more mitochondrial these peripheral membrane proteins become, the greater likelihood of them binding directly to golgin-97[14]. SPRYD7 is an example of a protein of unknown function that localises endogenously to vesicles and fits this criterion (Figs. 1e, 3a)[47].

An alternative explanation for our enrichment of adaptor proteins could be that golgin-97-mito recruits endosomes or TGN-derived vesicles to the mitochondria[48]. Lysosomal hydrolases are key markers of anterograde transport that traffic through the TGN and endosomes enroute to the lysosome. We did not observe any noticeable shift of lysosomal hydrolases to the mitochondria by golgin-97-mito in our analysis (Supplementary Fig. 3). Likewise, endosomal markers EEA1 and Rab5 did not show a detectable shift to mitochondrial fractions (Supplementary Data 2). We also performed correlative light and electron tomography (CLEM) on cells expressing golgin-97-mito and found an accumulation of spherical and oval structures surrounding the mitochondria that were roughly 30 to 80 nm in diameter, which correlates well with the predicted size of transport vesicles (Fig. 4a-e and Supplementary Movie 1)[49]. Together, these data support a model in which golgin-97-mito captures retrograde transport carriers formed by AP-1, SNX3, SNX4 and SNX-BAR proteins. It also suggests that a subset of adaptor proteins remain on endosome-to-Golgi carriers as they voyage to the TGN.

**Redirection of peroxisomes by mitochondrial golgins**. The ectopic tethering of large numbers of vesicles to golgin-coated mitochondria is likely to have some indirect effects on the rest of the cell. This is evident by the change in morphology of the mitochondria by electron microscopy as they become

increasingly zippered together by vesicles[11]. We detected a total of 43 proteins affected by golgin-97-mito that are not associated with the endosomal network (Supplementary Fig. 4a, b). Of these, 34 were peroxisomal proteins which were shifted to a more mitochondrial profile by both golgin-97-mito and GCC88-mito (Supplementary Fig. 4c). The extent of this shift was more pronounced with the former, which correlated with its stronger overall ability to capture vesicles (Fig. 1g). Consistent with this, peroxisomes were found adjacent to mitochondria in the presence of golgin-97-mito, but not in the presence of the BioID-mito control, when examined by immunofluorescence (Supplementary Fig. 4d). The reasons for this are unclear, but peroxisomes and mitochondria have a co-dependent relationship in the β-oxidation of fatty acids and the detoxification of reactive oxygen species[50]. Moreover, evidence of peroxisome-mitochondria contact sites, particularly at mitochondria-ER junctions, have been reported in yeast and mammalian cells[51–53]. Our LOPIT-DC data may provide further evidence of this co-dependence, but irrespective of the reasons it demonstrates the overall power and sensitivity of the LOPIT-DC coupled to Bayesian analysis approach.

**GCC88 and golgin-97 share a distinct pool of TMEM87A and TGN46 enriched vesicles**. Of 16 total proteins most strongly affected by GCC88-mito, the largest shifts in localisation were for TMEM87A and TGN46 (Fig. 1f). TMEM87A is a member of the enigmatic LU7TM family of GPCR-related proteins that has been implicated in endosome-to-Golgi retrograde transport and that localise to the TGN[54–56]. TGN46 and TMEM87A have nearly identical profiles in the control cells, and both were clearly shifted toward the mitochondria by GCC88-mito (Fig. 5a-c). As with TGN46, TMEM87A was also relocated to the mitochondria by golgin-97-mito (Figs. 5b, c, 2b). Furthermore, TMEM87A-RFP localised to spherical structures of roughly 50 nm diameter tethered adjacent to the mitochondria by golgin-97-mito analysed with CLEM (Figs. 4b–d, 5d)[49]. As described above, some cargo proteins relocated by golgin-97 are not captured by GCC88 and so TMEM87A must be enriched in a distinct class of retrograde transport vesicles.

The retrograde trafficking of IGF2R and M6PR from endosomes to the Golgi involves clathrin/AP-1, retromer-dependent and independent pathways[34,45,57,58]. However, our LOPIT-DC of GCC88-mito had no detectable effect on either of these cargo (Fig. 5b), suggesting that the distinct pool of TMEM87A and TGN46 enriched vesicles that GCC88 captures is not dependent on these particular retrograde pathways. GCC88-mito has been reported previously to capture IGF2R and M6PR enriched vesicles when expressed by transient transfection[13,58]. It is likely that the lower, stable, expression levels of GCC88-mito used in our LOPIT-DC experiments allowed us to only detect a distinct pool of vesicles.

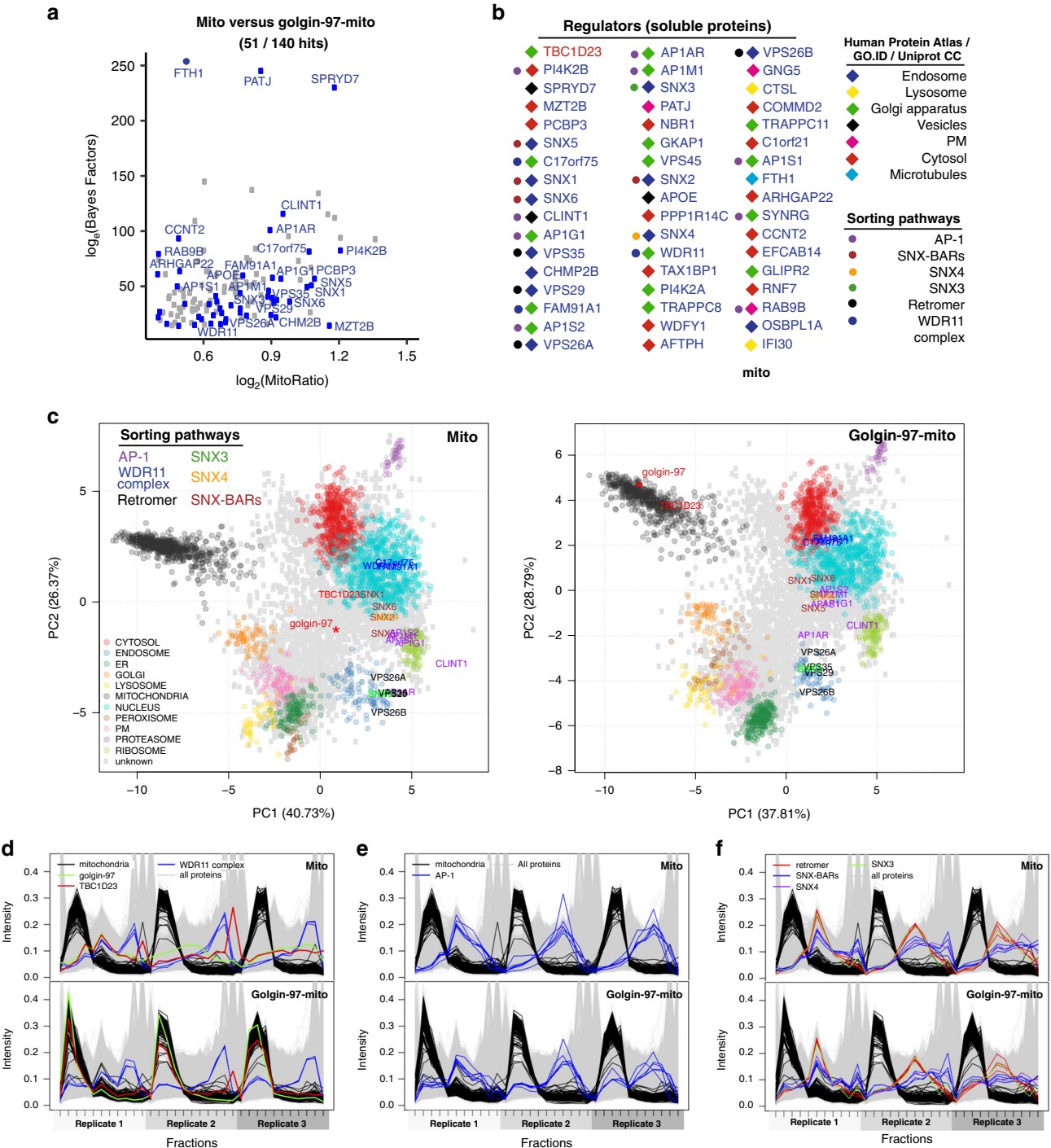

**Fig. 3 Adaptor proteins and accessory factors of vesicles specific to golgin-97. a** MitoRatio versus Bayes Factor analysis of mito versus golgin-97-mito showing all hits (blue) that are both localised to the endosomal network and are also soluble proteins (axes truncated as in Fig. 2a). **b** All hits shown in **a** ordered from left column to right column based on descending MitoRatio scores. Classification of their localisation and sorting pathway is based on multiple localisation databases and literature. **c** PCA projections for the LOPIT-DC of mito versus golgin-97-mito showing SVM classifications and the indicated proteins coloured based on the sorting pathway they have been proposed to belong to **a**, **b**. **d-f**. TMT reporter ion distributions of proteins across fractions for each replicate for mito versus golgin-97-mito showing the profiles of known mitochondrial markers (mitochondria) and all hits from **a**, **b** that have been reported belong to the indicated sorting pathways.

TGN46 has previously been shown to depend on golgin-97, golgin-245 and TBC1D23 for Golgi accumulation[14]. To extend this study to GCC88, we generated $\Delta gcc88$ mutants using CRISPR-Cas9 (Fig. 6a). When TGN46 is missorted from endosomes, it is diverted to lysosomes and degraded, allowing efficacy of its sorting to be quantified by both immunofluorescence and blotting[59], but

we observed only a mild decrease in TGN46 levels in the $\Delta gcc88$ mutant by blotting (Fig. 6a). Similarly, the localisation of TMEM87A to the TGN became more diffuse in $\Delta tbc1d23$ mutants and in $\Delta golgin-97$ / $\Delta golgin-245$ mutants by immunofluorescence, whereas the $\Delta gcc88$ mutant did not have a significant effect (Fig. 6b, c). Taken together, these results suggest that golgin-97

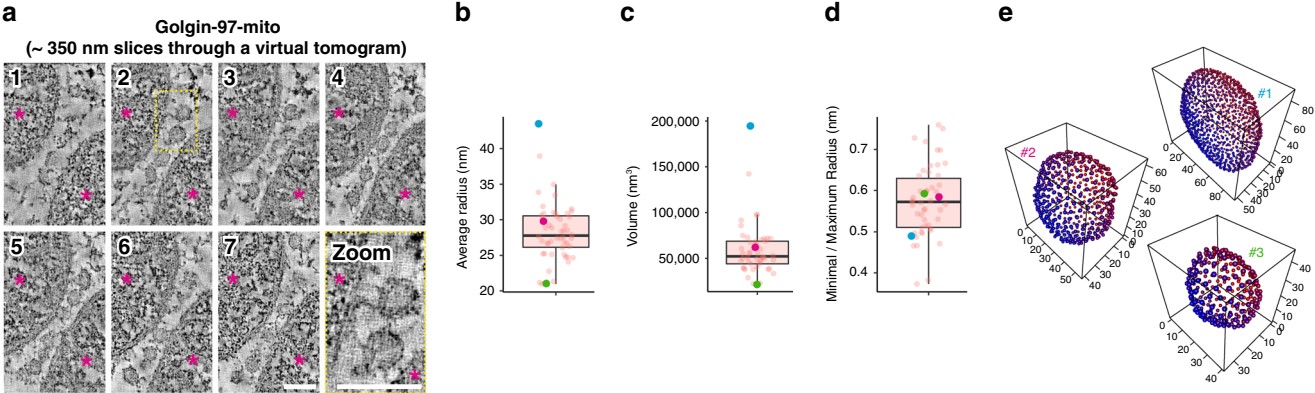

**Fig. 4 Electron tomography of vesicles captured by golgin-97-mito. a** CLEM of a Lowicryl-embedded HeLa cell expressing golgin-97-mito and TMEM87A-RFP throughout a thickness of approximately 350 nm. Mitochondria are large dense structures (pink stars) sandwiching small circular structures. Images are representative of two independent experiments. Scale bars 100 nm. **b–d** Boxplots showing quantitation of spherical structures in **a** through 3D segmentation using LimeSeg (50 quantified structures over two independent tomograms, data provided in the Source Data file), boxplots as for Fig. 1g. **e** 3D projections of indicated quantified structures in **b–d**, scale in nm. Data are provided as a Source Data file.

and golgin-245 also recruit TMEM87A and TGN46 enriched vesicles via TBC1D23, and are more efficient at capturing these vesicles than GCC88. Nonetheless GCC88 is still able to capture a set of vesicles that contains TMEM87A and TGN46 but not abundant retrograde cargo such as the mannose 6-phosphate receptors. Our results also indicate that GCC88 is redundant with golgin-97 and golgin-245 in capturing certain pools of vesicles. In support of this, knockdown of GCC88 in Δgolgin-97 / Δgolgin-245 mutants stopped cell growth (Fig. 6d).

In conclusion, we have illustrated the power of combining vesicle relocalisation with high-resolution spatial proteomics as an effective means of characterising the content of intracellular transport vesicles. Our study provides an extensive list of cargo proteins that move along these retrograde routes, as well as peripheral membrane proteins that may act on these carriers, which will hopefully provide a valuable resource for further investigation on this important aspect of intracellular membrane traffic. We have validated four cargo that had not previously been reported as being captured by mitochondrial golgins (ATG9A, TVP23B, furin and TMEM87A). Nonetheless, it should be noted that we applied an empirical cut-off to select a set of high scoring hits for further analysis and, like any statistical cut-off, it is to some extent arbitrary. Thus, some of the other hits may not be in vesicles, and conversely some of those below the cut-off, especially those close to the boundary, may actually be in vesicles. However, the full list of data provided in Supplementary Data 2, and the relative simplicity of the mitochondrial golgin relocation assay should enable others to readily test the vesicle location of their proteins of interest. Finally, this approach is not limited to trafficking vesicles and could be applied to any protein complex or sub-cellular compartment with resident proteins that can be relocated to an ectopic site. Thus, we present a general strategy to define intracellular sub-proteomes within their native cellular environment.

## Methods

**Plasmids and antibodies**. The golgin-mito plasmids used for transient transfection were generated from C-terminally truncated golgins that were PCR-amplified from human GCC88 and golgin-97 cDNA using GCC88_F / GCC88ΔC_R and G97_F / G97ΔC_R primers (Supplementary Table 1), and cloned into a pcDNA3.1 + plasmid so as to express the entire golgin without the GRIP domain followed by an HA epitope tag and the mitochondrial-targeting TMD of monoamine oxidase (MAO)[13]. GCC88-mito, golgin-97-mito and Mito were PCR-amplified from these plasmids with GCC88_F / Mito_R, G97_F / Mito_R and Mito_F / Mito_R, and cloned into pcDNA5/FRT/TO to generate pMX0269, pMX0267 and pMX0266 used to make stable lines. BioID was PCR-amplified using BioIDC' F and R primers

and subcloned into pcDNA3.1+ and pcDNA5/FRT/TO plasmids tagged to Mito to generate BioID-mito used for transient transfection (pJJS112) and to make stable lines (pJJS128). Furin-GFP (pJJS084) for transient transfection was generated by PCR-amplification of human Furin CCDS10364.1 cDNA using FurinC' F and R primers, and cloned into a pcDNA3.1+ plasmid tagged to EGFP. TMEM87A-RFP (pJCO003) for transient transfection was generated from PCR-amplification of TMEM87A from 293 cells using TMEM87AC' F and R primers, and cloned into a pcDNA3.1+ plasmid tagged to mRFP. eSpCas9 (Addgene, 71814) and pIRESpuro3 (Clontech) were used to generate CRISPR-Cas9 stable cell lines. eSpCas9 was digested with BbsI and GCC88cas9, G97cas9, G245cas9, TBC1D23cas9 Top and Bottom primers were annealed and inserted by oligonucleotide cloning to generate plasmids used to make Δgolgin-97, Δgcc88, Δgolgin-245, Δtbc1d23 mutant cell lines[14]. All primary antibodies used in this study are described in Supplementary Table 2. Secondary antibodies were Alexa Fluor-conjugated donkey sera against the relevant species (Thermo Scientific).

**Cell culture, transfection, CRISPR knockout cell lines and immunofluorescence**. Cell lines HeLa (ATCC) and Flp-In T-REx 293 (Thermo Scientific, R78007) were cultured in Dulbecco's modified Eagle's medium (DMEM; Invitrogen) supplemented with 10% foetal calf serum (FCS) and penicillin/streptomycin at 37 °C and 5% $CO_2$. To transduce Flp-In 293 cells with pMX0267, pmX0269, pMX0266 and pJJS128, cells were grown in 6-well plates to ∼50% confluence and transfected with 1 µg pOG44 (Flp recombinase vector) and 1 µg pcDNA5/FRT/TO-based plasmid using 6 µl FuGENE 6 in 200 µl of Opti-MEM. The medium was replaced after one day, and after two days cells were trypsinised, expanded in medium containing 100 µg ml⁻¹ hygromycin, and verified by induction of expression with 1 µg ml⁻¹ doxycycline and immunoblotting for the HA epitope. Plasmid pJJ269 was used to generate Δgcc88 Hela cells as for the Δtbc1d23 mutant and Δgolgin-97 / Δgolgin-245 mutant HeLa cells[14]. All cell lines were tested regularly to ensure that they were mycoplasma free (MycoAlert, Lonza). For immunofluorescence, cells were transfected with plasmid DNA using FuGENE 6 according to the manufacturer's instructions (Promega). Cells were fixed with 4% formaldehyde in PBS and permeabilised in 0.5% (v/v) Triton X-100 in PBS. Cells were blocked for one hour in PBS containing 20% FCS and 0.25% Tween-20. If this protocol did not show good Golgi staining for a particular antibody, cells were permeabilised and blocked in 20% FCS and 0.1% saponin. Primary and secondary antibodies were applied sequentially in blocking buffer at dilutions given in Supplementary Table 2. For quantitation of immunofluorescence experiments, a cell mask was defined by over-exposed DAPI. HA and GM130 were used to mark the mito construct and the Golgi, respectively (NIS-Elements, Nikon). Then the mean intensity of the cargo channel in the mitochondria mask (excluding the areas that overlap with the Golgi mask) were divided by the mean intensity of the cargo channel of the Golgi mask for each quantified cell. The mitochondria and Golgi mask had a minimum threshold of 18% of the mean labelling of their respective markers to exclude poorly expressed constructs or out-of-focus Golgi. For quantification of Golgi localisation, the mean intensity of the cargo in the Golgi mask was divided by the mean intensity of GM130 in the Golgi mask.

**Cell viability assay**. Δgolgin-97 / Δgolgin-245 HeLa cells were cultured in DMEM supplemented with 10% foetal calf serum (FCS) at 37 °C and 5% $CO_2$. On day 0, a 6-well plate was seeded with 180,000 cells per well to obtain a confluency of 40–50% the following day. Cells were transfected in triplicate on day 1 and day 5 with 30 pmol siRNA against GCC88 (Dharmacon ON-TARGETplus human GCC1 SMARTpool L-017478-00-005) or control RNAi (Dharmacon ON-TARGETplus

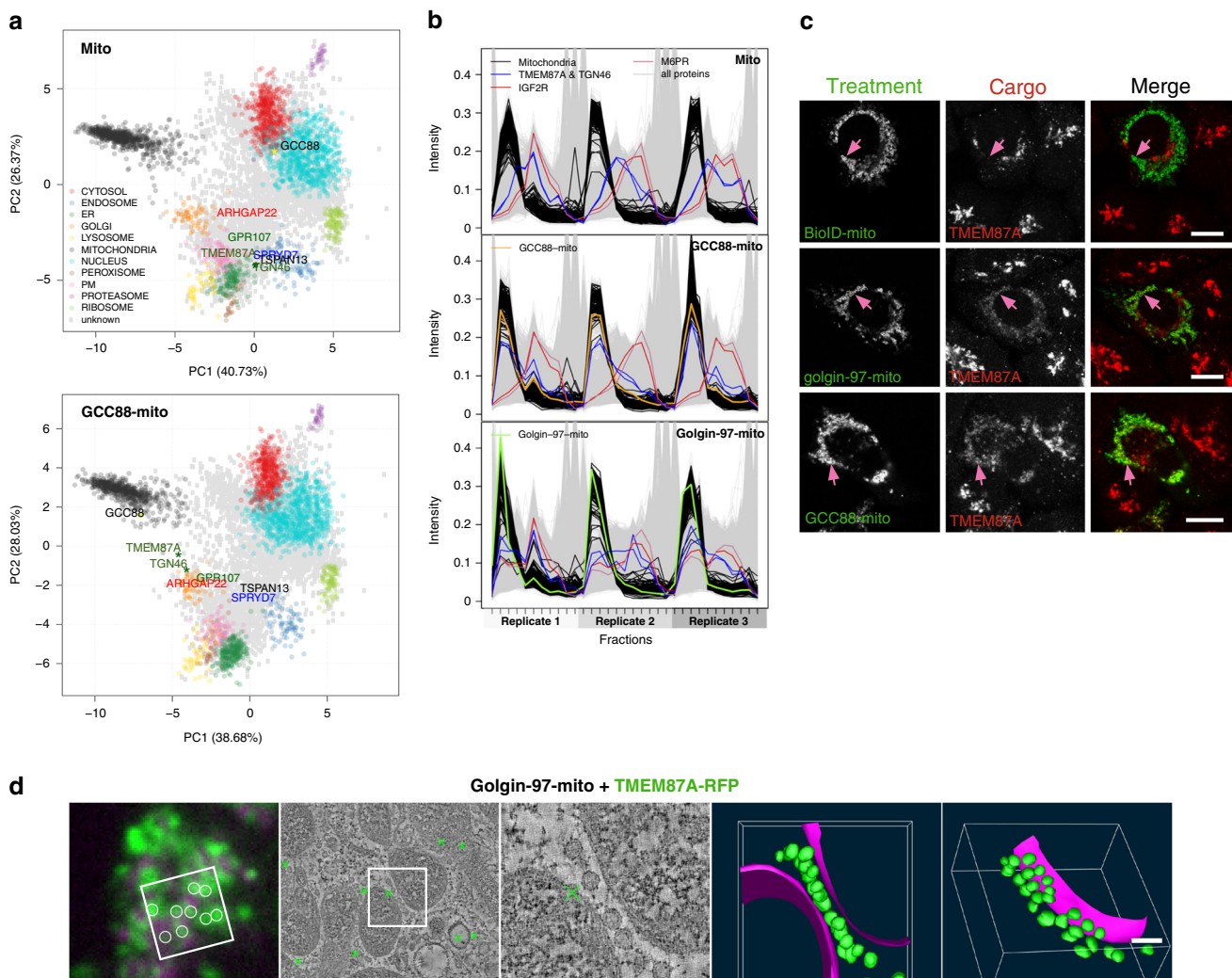

**Fig. 5 GCC88 and golgin-97 share a distinct pool of TMEM87A and TGN46 enriched vesicles. a** PCA projections for the LOPIT-DC of mito versus GCC88-mito showing SVM classifications and the top six hits that are localised to the endosomal network from Fig. 1f. **b** TMT reporter ion distributions of proteins across fractions for each replicate for mito, golgin-97-mito and GCC88-mito showing the profiles of known mitochondrial markers (Mitochondria) and the indicated proteins. **c** Confocal micrographs of HeLa cells expressing BioID-mito and the indicated golgin-mito constructs (HA) stained for endogenous TMEM87A (cargo). Micrographs representative of three independent experiments, scale bars 10 μm. **d** Correlative light and electron tomography of resin-embedded HeLa cells co-transfected for 24 hours with golgin-97-mito and TMEM87A-RFP (green), and stained with mitotracker DeepRed (magenta) to label mitochondria. Left: Fluorescence micrograph of a section of a resin-embedded cell. White circles indicate TMEM87A-RFP signals localised in the electron tomogram. Middle-left: Virtual slice from the electron tomogram acquired at the area indicated with a white square on the left image—green crosses indicate predicted positions of the TMEM87A-RFP signal centroids shown on the fluorescence image. Tomogram representative of two independent replicates. Middle: magnified area from white square in the tomogram slice showing vesicles accumulated between two mitochondria, with two views of a 3D segmentation model of this same area (mitochondria in purple and vesicles in green). Full tomogram and rendering in Supplementary Movie 1.

non-targeting pool D-001810-10-05) using 9 μl Lipofectamine RNAiMAX (Thermo Scientific) in a final volume of 250 μl Opti-MEM media as described in the manufacturer's instructions. Starting on day 2, and thereafter every 48 h, cells were washed twice in EDTA, trypsinised and resuspend in an appropriate volume of complete media and counted in a Countess II Automated Cell Counter (Thermo Scientific). Cells were subsequently either re-plated in their entirety (GCC1 RNAi) or diluted so that they would not overgrow during the course of the experiment (control RNAi), with a correction factor applied to the cell count to take this dilution into account.

**Immunoblotting**. One 75 cm² flask of cells at ~90% confluence was harvested by scraping and centrifugation (1000 × g, 5 min), washed twice with ice-cold PBS, resuspended in 300 to 500 μl lysis buffer (50 mM Tris pH 7.4, 0.1 M NaCl, 1 mM EDTA, 1% Triton X-100, 1 mM PMSF, cOmplete inhibitors) for 30 min on ice, clarified by centrifugation (10,000 × g, 10 min at 4 °C), and protein concentration was determined (BCA assay, Thermo Scientific). Lysate concentrations were

normalised, and the lysates were heated in NuPAGE SDS sample buffer containing 10% β-mercaptoethanol at 90 °C for 5 min, run on a gel and transferred to nitrocellulose. All blots were blocked in 5% (w/v) milk in PBS-T (PBS with 0.1% (v/v) Tween-20) for 1 h, incubated overnight at 4 °C with primary antibody in the same blocking solution (Supplementary Table 2), washed three times with PBS-T for 5 min, incubated with a 1:3000 dilution of species-specific HRP-conjugated secondary antibody (GE Life Sciences) in 0.1% (v/v) milk in PBS-T for 1 h, washed five times with PBS, and detected with Immobilon Western HRP substrate (Merck).

**Correlative fluorescence microscopy and electron tomography of resin-embedded cells**. Correlative fluorescence microscopy and electron tomography (CLEM) of resin-embedded cells used established methods as follows[49,60]. HeLa cells were grown on carbon-coated 3 mm sapphire discs (Wohlwend GmbH) in six-well plates for 24 hours and transfected with 1 μg golgin-97-mito and 1 μg TMEM87A-RFP using FuGENE 6, and stained with MitoTracker Deep Red. The

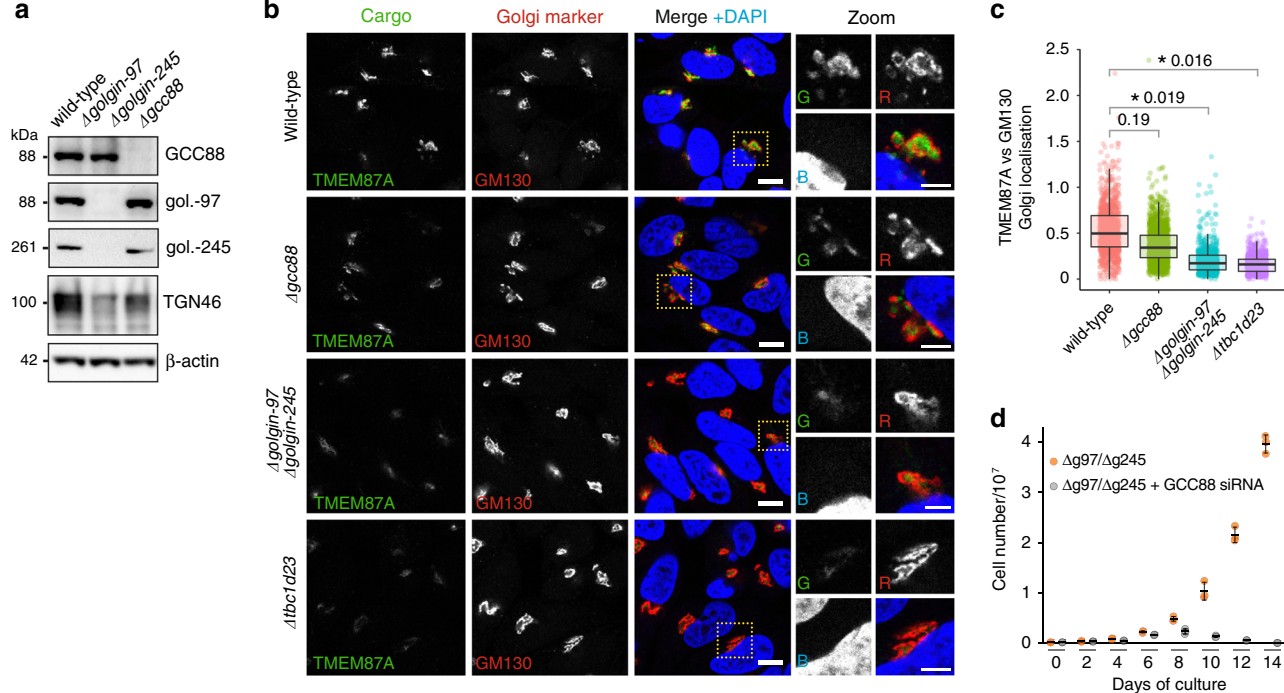

**Fig. 6 GCC88 and golgin-97/golgin-245 have redundant roles that together support cell viability. a** Immunoblot of proteins from wild-type and mutant HeLa cells probed with antibodies against the indicated proteins, blot representative of two independent experiments. **b** Confocal micrographs of wild-type and mutant HeLa cells stained for endogenous TMEM87A (cargo), GM130 (marker of *cis*-Golgi) and DAPI. Micrographs representative of three independent experiments, scale bars, 10 μm (5 μm in zoom). **c** Quantification of **b** measuring the ratio of the intensity of TMEM87A at the Golgi over that of GM130. Boxplots are of data from three independent replicates, $n \geq 300$ per replicate, as for Fig. 1g. P-values are of two-sided two-sample *t*-tests comparing the means of ratios from the three replicates for each cell type. **d** Effect of knockdown of GCC88 on growth of Δgolgin-97 / Δgolgin-245 double knockout HeLa cells - three replicates per time point (error bars show mean and SD). Data are provided as a Source Data file.

cells were frozen under high pressure using an HPM100 (Leica Microsystems), freeze-substituted using 0.008% uranyl acetate in acetone and embedded in Low-icryl HM20 (Polysciences) using an AFS2 (Leica Microsystems). The samples were sectioned into 350 nm thin sections using a microtome (Leica Microsystems) and a diamond knife (Diatome). The sections were collected on 200 mesh copper grids with carbon support (S160, Agar Scientific Ltd). 100 nm TetraSpeck microspheres (Invitrogen, T7279) diluted 1:200 in PBS pH 8.4 were adsorbed for 5 min to the sections and used as fiducial markers for correlation. Fluorescence images were acquired using a Ti2 wide field microscope (Nikon) equipped with a Niji LED light source (Bluebox Optics), a ×100 oil-immersion TIRF objective (NA = 1.49) and a Neo sCMOS DC-152Q-C00-Fl camera (Andor). Filters: 49002 ET-GFP (Chroma), 49005 ET-DSRed (Chroma), 49006 ET-Cy5 (Chroma). Electron microscopy was done using a Tecnai F20 (FEI) operated at 200 kV and a high tilt tomography holder (Fischione Instruments; Model 2020). Montage images of regions of interest were acquired on a BM-Orius detector using transmission EM at approximately 130 μm defocus, at a pixel size of 1.1 nm and using the montaging function in SerialEM v3.8.0[61]. The correlation between the montage images and the fluorescence images was achieved by matching the fiducial marker positions in the EM image and the fluorescence image[60,62]. Electron tomographic tilt series were acquired from 60 to −60 degrees with one degree increments, using the Scanning TEM mode (STEM) on an axial bright field detector, at pixel sizes of 1.1 nm and 2.2 nm. Tomograms were reconstructed using IMOD v4.10.29[63]. The dataset is representative of 18 correlated tomograms were acquired from three cells distributed on two sapphire discs. Circular structures approximately 100 nm adjacent to mitochondria in tomograms were 3D segmented using the ImageJ plugin LimeSeg v0.4.4[64]. The average, minimal and maximal radius, along with overall volume of the 3D segmentations of each structure were then measured. The 3D segmentation model was made by manual tracing followed by simplification and smoothening using Amira v20.19.4 (Thermo Scientific) and IMOD v4.10.29[63].

**LOPIT-DC sample preparation**. Organelle proteomics of Flp-In 293 lines expressing inducible mito-golgins was performed using the LOPIT-DC method[23]. Cells from at least eight T175 flasks per treatment were passaged so that they were approximately 50% confluent after 24 h. After this time, 1 μg ml-1 doxycycline was added to the medium and incubated for 48 h. The cells were pelleted at 200 × g and washed thrice in ice-cold PBS, and then weighed to measure cell quantity. A minimum of 0.7 g of cell pellet were resuspended in 4 ml of ice-cold detergent-free lysis buffer (LB) (250 mM sucrose, 10 mM HEPES pH 7.5, 2 mM EDTA, 2 mM

magnesium acetate tetrahydrate, cOmplete protease tablet (Roche)) and incubated on ice for 5 min. The cell suspension was then lysed with a ball bearing homogeniser using a 12 μm tungsten carbide ball (Isobiotec) until approximately 90% of cells were visibly lysed by trypan blue staining. Unlysed cells and insoluble cell debris were pelleted from the homogenate by centrifugation at 200 × g for 20 min three times. A nuclei prep (NUC) was then performed on the collected pellet by resuspending it in 25% OptiPrep (Sigma-Aldrich), underlaying this with 30% and 35% OptiPrep, and then centrifuging this gradient at 10,000 × g for 20 min at 4 °C. The interface containing the nuclei was carefully collected with a syringe, washed 6 times with LB by pelleting at 22,000 × g for 20 min each, and the final NUC pellet then frozen. The cell homogenate was then fractionated by differential ultra-centrifugation through the following sequential spins at 4 °C: P1 pellet, 1000 × g for 20 min; P2 pellet, 3000 × g for 10 min; P3 pellet, 5000 × g for 10 min; P4 pellet, 9000 × g for 15 min; P5 pellet, 12,000 × g for 15 min; P6 pellet, 15,000 × g for 15 min; P7 pellet, 30,000 × g for 20 min; P8 pellet, 79,000 × g for 43 min; P9 pellet, 120,000 × g for 3 h. Each collected pellet was spun again under the same condition and the final pellet was frozen. The pellets were thawed the next day and both the NUC and P1 pellet were treated with 250 units of benzonase nuclease (Sigma-Aldrich, E1014) for 10 min on ice to reduce viscosity. All pellets were then solubilised with approximately 200 μL of membrane solubilisation buffer (MSB) (50 mM HEPES pH 8.5, 0.2% SDS, 8 M Urea) and sonicated for 3 times 30 s pulses on highest setting while avoiding overheating above room temperature. Solubilised fractions were then clarified by centrifugation at 20,000 × g for 5 min. The protein concentration of each fraction was measured using the Pierce BCA Protein Assay Kit, and a 50 μg per 100 μL aliquot of each fraction was then reduced by incubating with 10 mM DTT for 2 h at room temperature, and alkylated by incubating with 25 mM iodoacetamide (Sigma-Aldrich, I6125) for 2 h at room temperature. The samples were then diluted with 1 volumes of water, and precipitated in at least 6 volumes of pre-chilled acetone overnight at −20 °C. The precipitated pellets were decanted and allowed to air dry for 3 min, resuspended in 100 μL of 50 mM HEPES pH 8.5 buffer, vortexed, sonicated and then trypsinised overnight according to the manufacturer's instructions (Promega, V5111). The subsequent peptide fractions were labelled with TMT10plex and TMT11plex (Thermo Scientific), multiplexed according to the manufacturer's instructions, and then lyophilised. The samples were then resuspended in 0.8 mL 0.1% trifluoroacetic acid (Thermo Scientific, 85183) and desalted with Sep-Pak tC18 (Waters, WAT036820), and then lyophilised. The samples were resuspended in 100 μL of 95% Buffer A (20 mM ammonium formate, pH 10), 5% Buffer B (20 mM ammonium formate pH 10, 80% acetonitrile). High-pH reverse phase chromatography was performed using a

Waters Acquity UPLC with gradients of Buffer A and Buffer B through Acquity BEH C18 column and trap column (Waters, 186002353 and 186003975), and collected into a series of pre-fractions. Only pre-fractions found within a Gaussian distribution of peptides determined by the UV detector were kept. The first and middle pre-fractions were concatenated, and this pattern was repeated with every subsequent pre-fraction until 18 total concatenated pre-fractions remained and were lyophilised.

**Mass spectrometry**. Liquid chromatography was performed on the concatenated RP-UPLC pre-fractionated TMT-labelled peptides on a fully automated Ultimate 3000 RSLC nano System (Thermo Scientific) fitted with a 100 μm × 2 cm Pep-Map100 C18 nano trap column (Aclaim PepMap, Thermo Scientific) and a 75 μm × 25 cm, nanoEase C18 T3 column (Waters). Samples were separated using a binary gradient consisting of buffer A (2% MeCN, 0.1% formic acid) and buffer B (80% MeCN, 0.1% formic acid) with a flow rate of 300 nL/min. The HPLC system was coupled to a Q Exactive Plus mass spectrometer (Thermo Scientific) equipped with a nanospray ion source. The mass spectrometer was operated in standard data-dependent mode, with MS full-scan at 380–1600 $m/z$ range, and a resolution of 70,000. This was followed by MS2 acquisitions of the 15 most intense ions with a resolution of 35000 and NCE of 33%. MS target values of 3e6 and MS2 target values of 1e5 were used. Isolation window of precursor was set at 0.7 Da, and dynamic exclusion of sequenced peptides was enabled for 40 s.

**LOPIT-DC data processing**. Raw data files were processed using Proteome Discoverer v2.1 (Thermo Scientific). For the processing work flow, reporter ions were scanned based on HCD activation, MS2 order, FTMS mass analyser with collision energy between 0 and 1000. The peak integration method was set to the most confident centroid with a 2 milli mass unit integration tolerance. Precursors were selected based on MS1 with a precursor mass between 350 and 5000 Da and minimum peak count of 1. Scan event filters were set to MS2 order, collision energy between 0 and 1000, with full scan type. Peak Filters of S/N Threshold were set to 1.5. The Mascot server was searched against the Reviewed (Swiss-Prot) Human Proteome (from UniProtKB at the UniProt database) with precursor and fragment ion tolerances of ±20 ppm and ±0.2 Da, and up to two missed tryptic cleavages permitted. Static modifications were set as carbamidomethylation of cysteine and TMT 6-plex modification of lysine and peptide N termini. Dynamic modifications were set as acetyl of the protein N-termini and oxidation of methionine. The PSMs for the 'forward' and 'decoy' searches by Mascot were re-scored using the Percolator algorithm to yield a more robust false discovery rate. For the consensus work flow, protein grouping was enabled, with peptides used based on 'Unique + Razor'. Reporter quantification was based on an average reporter S/N of 5 and co-isolation threshold of 30. No normalisation or scaling were applied. Quantifications were hypothesis tested with ANOVA for individual proteins, a maximum fold change of 100, and the ratio was calculated using the summed abundance. Peptide group modifications were set to a site probability threshold of 75. Peptides were validated using the 'automatic control peptide level rate if possible' option using a strict and relaxed target false discovery rates (FDR) of 0.01 and 0.05. Peptides were then filtered for High FDR and master protein with a minimum peptide length of 5. No imputation was performed. To improve signal-to-noise, proteins with a CRAPome FREQ ≥ 0.1 (Supplementary Data 1)[65], were filtered out prior to analysis unless they were organelle markers use for SVM and TAGM classifications, along with keratins, serum albumin and trypsin[66]. The data were then used to generate spatial maps containing SVM and TAGM classifications using the Bioconductor v3.10 pRoloc work flow (see below)[67]. SVM classifications were set to an FDR of 0.05 based on the consensus from sub-cellular localisations obtained from the Human Protein Atlas and Uniprot using the BioMart portal [https://www.ensembl.org/info/data/biomart/index.html][68]. All LOPIT-DC experiments were repeated three times on separate days, with the treatments of each replicate performed concurrently on the same day.

**Posterior localisation probabilities**. The posterior probability that a protein belongs to a sub-cellular niche, henceforth referred to as the posterior localisation probability, is computed using the TAGM-MAP method[25]. Briefly, the parameters of the T-augmented Gaussian mixture model are learnt by maximising the log posterior of the parameters with respect to the data, to obtain *maximum a posteriori* (MAP) estimates of the parameters. The posterior localisation probability that a protein belongs to each organelle is then computed and the most probable sub-cellular niche is reported. Prior choices were made using the default settings in the pRoloc work flow for the Bayesian analysis of spatial proteomics in the R (v3.6) package Bioconductor (v3.10)[67]. Proteins predicted to localise to the mitochondria and nucleus were discarded from subsequent analysis.

**Bayesian non-parametric two-sample test**. To detect perturbations in the quantitative protein profiles, we apply a Bayesian non-parametric two-sample test[69]. First, the data are transformed using the additive log ratio transform[70]. We then proceed to test whether the protein profiles are different between the control and treatment. Formally, we test against two contrasting models. The first model posits that the quantitative protein profiles in each experiment (control and treatment) are drawn from an identical shared distribution. Whilst the second supposes that there are

independent models for each of the control and treatment. The log Bayes factor is used to objectively determine support for one model over the other, where larger log Bayes factors are considered support for the independent model[71]. A non-parametric prior over functions, the Gaussian process, is specified with squared exponential covariance[72]. Default Gamma priors were used for the hyperparameters of the Gaussian process[69]. The natural logarithm of the Bayes factors is reported.

**Mitochondrial ratios**. To determine the biological relevance of quantitative protein profiles that are shifted between treatment and control, we computed a mitochondrial ratio (MitoRatio) as follows. For each protein in each experiment, the squared Mahalanobis distance[73] to the mean of the quantitative profiles of the mitochondrial marker proteins is computed, where a robust estimate for the covariance is used[74]. Then, for each protein, the ratio of the distances in control and treatment is computed, proceeded by a $\log_2$ transform. Proteins that move closer to the mitochondria upon the treatment have larger mitochondrial ratios.

**Reporting summary**. Further information on research design is available in the Nature Research Reporting Summary linked to this article.

## Data availability

Mass spectrometry data used in this study are summarised in Supplementary Data files 1 and 2. All protein-level datasets generated during this study are available in the Bioconductor pRolocdata package (version ≥ 1.25.2) at https://github.com/lgatto/pRolocdata. Interactive versions of the PCA plots can be viewed online through dedicated R Shiny apps at https://proteome.shinyapps.io/golgins_mito/, https://proteome.shinyapps.io/golgins_golgin97mito/ and https://proteome.shinyapps.io/golgins_gcc88mito/. The mass spectrometry proteomics data have been deposited to the ProteomeXchange Consortium via the PRIDE[75] partner repository with the dataset identifier PXD018110. Protein localisation data from UniProt [https://www.uniprot.org/help/subcellular_location] and the Human Protein Atlas [https://www.proteinatlas.org/] were obtained using the BioMart portal [https://www.ensembl.org/info/data/biomart/index.html][68]. All reagents generated by this study are available from the corresponding authors on request. Source data are provided with this paper.

## Code availability

No custom code was used to generate, test or process the data described herein. Peptide spectrum matching and quantification to protein-level abundances were performed in Proteome Discoverer v2.1(Thermo Scientific) as described in the LOPIT-DC data processing section. Protein localisation analyses were performed using the freely and openly available R package Bioconductor (v3.10) using pRoloc as described in the LOPIT-DC data processing section, and the Posterior localisation probabilities section [http://bioconductor.org/packages/release/bioc/html/pRoloc.html][67].

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

## Acknowledgements

The authors thank Manu Hegde for comments on the manuscript, Josie Christopher for help with LOPIT-DC, Owen Vennard for help with multimedia in figures, Wanda Kukulski, Alejandro Melero-Carrillo and Jérôme Boulanger for assistance in analysing the CLEM data, Mike Deery and Yangesh Umrania of the Cambridge Centre for Proteomics, and the members of the LMB Mass Spectrometry Facility. Funding was from the Medical Research Council (MRC file reference number MC_U105178783). J.J.H.S. was supported by a European Molecular Biology Organisation long-term fellowship, L.B. by European Union Horizon 2020 INFRAIA project Epic-XS (project no. 823839), and O.M.C by a Wellcome Trust Mathematical Genomics and Medicine studentship.

## Author contributions

J.J.H.S. and S.M. devised, and J.J.H.S., S.M. and K.S.L. planned, the study. J.J.H.S. performed the LOPIT-DC experiments. J.J.H.S. and S.Y.P.C performed the mass spectrometry. J.J.H.S. and O.M.C. analysed the LOPIT-DC data, and L.B. constructed the Shiny apps. J.J.H.S. and J.C.O. performed the immunofluorescence and blotting experiments, and A.G. tested cell viability. A.B. and J.C. performed the electron tomography. J.J.H.S. and S.M. wrote the manuscript.

## Competing interests

The authors declare no competing interests.
