## [Peer Review File · Nature Communications]

Reviewers' comments:

Reviewer #1 (Remarks to the Author):

This manuscript describes the application of spatial proteomics to identify constituents of the transport vesicles mediating endosome to Golgi trafficking. The authors use an in-cellulo vesicle-tethering assay in which golgin vesicle tethering proteins are ectopically localized to mitochondria to capture transport vesicles arriving from endosomes. This is combined with subcellular fractionation and quantitative mass spectrometry to identify those vesicle proteins selectively enriched or shifted towards the mitochondrial fraction, which correspond to constituents of the tethered vesicles. There are differing degrees of enrichment of different vesicle proteins, consistent with different golgins capturing distinct classes of vesicles.

The data are of high quality and overall I found the work to be very convincing. The approach is highly innovative and may be applied to the study of other vesicle trafficking steps. Until now, it has been a major challenge to identify the content of transport vesicles, and the approach shown here indicates that it is feasible, and has identified the complements of endosome to Golgi vesicles with a high degree of confidence. This is an important advance. The study is likely to be of broad interest and I strongly support its publication. There are a few issues that the authors should consider though.

- 1.) The text mentions 16 proteins rerouted to the GCC88 mitochondria, but, unless I missed it, they are not all named (I could only see six vesicle proteins in figure 4a). As is done for golgin-97, the entire complement of GCC88-tethered vesicle proteins should be listed.
- 2.) The supplementary table is very difficult to follow. It should be reconfigured to make it more legible as to what the different columns and numbers are.
- 3.) To exclude the possibility that endosomes are not stuck to the golgin-coated mitochondria, lysosomal hydrolases are studied. The data are consistent with these not being rerouted to the golgin, but it would also be good to mention more typical endosomal markers in this respect e.g. EEA1, Rab5. Presumably these are not shifted to the mitochondrial fractions? If so, it would be good to mention.
- 4.) To make the tomogram (supplementary video) easier to visualize, it would be good to make a 3D reconstruction, showing the volume of the mitochondrion with the vesicles adhered to the surface.
- 5.) Based upon the Western blot shown in Fig 4h, it is stated that combined KO/KD of all the 3 golgins studied causes reduced cell growth. Cell counting should be done to confirm that growth is indeed reduced.

Reviewer #2 (Remarks to the Author):

The manuscript entitled “Spatial proteomics defines the content of trafficking vesicles captured by golgin tethers” describes an application of LOPIT-DC to identify the content of two different endosome-derived vesicles using a strategy that capture them at the mitochondria. In this work, the authors used two proteins, golgin-97 and GCC88, to identify cargoes specific to these individual golgins. The paper thus provides an extensive list of cargo and membrane proteins, which includes both known and novel proteins, and support the use of this approach to identify proteins found in different subcellular organelle by tethering them to an ectopic site.

Overall, the manuscript is interesting and demonstrate a novel and elegant application of LOPIT. The data presented is very convincing, and the analysis is very thorough using supervised machine learning and Bayesian statistical modelling. However, the lack of validation of the proteins identified using mutants and by additional control experiments would be necessary to support a physiological relevance of the observed inventory of proteins to make sure that the identification of the protein profiles is not an artefact of targeting organelles to an abnormal destination. The KO presented in figure 4 appear to demonstrate redundancy in those pathways, and do not conclude whether the differences observed by LOPIT when those organelles are tethered to the mitochondria are actually representative of sorting in the golgi.

The threshold used for identifying proteins from the complex consist of the WDR11 complex. While I understand that this complex is a good indicator of proteins found, there could be other proteins with lower enrichment scores that could also be relevant. Considering the analysis presented and the triplicates, I would think that using a proper significant threshold would be more relevant then using known proteins. Are these proteins even considered significantly enriched in their analysis? Is this threshold actually more stringent, or are those proteins simply too close to background?

Other comments:

It is not clear why they used CRAPome to filter out some proteins? It is not explained in the text.

The supplementary table includes the dataset from the proteomic experiment with the intensity values, but there are no tables with the proteins enriched or grouped from their bioinformatics analysis.

Reviewer #3 (Remarks to the Author):

Comments on paper

Spatial proteomics defines the content of trafficking vesicles captured by golgin tethers

John J.H. Shin^{1*}, Oliver M. Crook^{2,3,4}, Alicia C. Borgeaud¹, Jérôme Cattin-Ortolá¹, Sew-Yeu, Peak-Chew¹, Jessica Chadwick¹, Kathryn S. Lilley^{2,3}, Sean Munro^{1*}

Previously, the authors showed that re-targeting of golgins to mitochondria is sufficient to also re-

distribute Golgi-directed trafficking vesicles (Mie Wong and Sean Munro 2014). Here, in order to unravel the protein content of these vesicles, the authors use innovative methods to isolate mitochondria together with the trapped, mis-localised vesicles followed by MS. Their approach opens interesting new possibilities in the determination of cargo proteins present in small, rare and transient trafficking vesicles that are otherwise difficult to isolate and study. The ultimate aim of these studies is to gain insight in cargo composition of vesicles carrying specific tethering proteins.

Detailed comments:

- As discussed in the text, mito-protein overexpression causes cell stress, which leads to changes in mitochondria morphology. My major worry of the experimental set up is whether the accumulation of some proteins/vesicles nearby the mitochondria could be an artefact due to mito-protein-stress. EM pictures of the negative controls (like overexpression of mito-Tag without golgin protein) would help to understand better the impact of this overexpression. Moreover, silencing of the retromer components should inhibit the recruitment to mitochondria. These controls would improve the sturdiness of the approach.
- The authors claim that there is a general shared result between golgin-97 and golgin-245. It seems to me that this is based only on the observation that both proteins bind AP-1 and TBC1D23. Experiments were done with golgin-97 only. The finding that golgin-97 and golgin-245 have some common interacting partners does not rule out that they could also be on different vesicles. The conclusions on golgin-245 are therefore a bit too speculative.
- The paper is mostly based on the results obtained for golgin-97-mito. There is much less on GCC-88-mito and authors were not able to determine the cargo proteins of these vesicles. Thus, this is a study mostly on golgin-97 rather than a comparison between cargo of vesicles with different tethers. This should be better reflected in the text and title.
- Figure 1a shows a model of the re-targeting system. However, this shows mito-golgin-97-BioID, whereas they refer to a paper where this fusion protein is made without BioID. The model should be made more compatible with the actual experimental set up.
- Figure 1h-i – authors show GCC185-mito staining but this is not cited in the result text.
- Figure 3g and Figure 4d.
Authors claim that cells are selected based on CLEM using fluorescent staining of TMEM87A-RFP and mitotracker. However, I miss here the golgin-97-mito staining. Unless 100% of the cells are co-transfected with TMEM87A-RFP and golgin-97-mito we cannot conclude from the present set of data if golgin-97-mito is indeed present on these mitochondria.
The EM figures show long stretched proteins in between the vesicles. Can the authors comment if these are actually represent the presence of tethers?
- PCR projection for LOPIT_DC: the pale coloured bullets are not easy to distinguish

Response to reviewers' comments.

We are pleased that the reviewers felt that the work was highly innovative and the data very convincing, and that they supported publication subject to addressing their points. We have added new analysis, data and text to address the reviewers' comments. These changes are described below, along with some additions to describe our arrangements for data sharing.

Data availability.

The mass spectrometry proteomics data have been deposited to the ProteomeXchange Consortium via the PRIDE partner repository with the dataset identifier PXD018110. This is available to reviewers with Username: reviewer28255@ebi.ac.uk and Password: hbeLw5Pj. We have also provided an interactive web app to visualise the location of a chosen protein in the spatial maps and in the gradient profiles. The links are listed in the Data Availability paragraph. We have assembled a source data file and this is mentioned in the figure legends and the Data Availability paragraph.

Reviewer #1 (Remarks to the Author):

This manuscript describes the application of spatial proteomics to identify constituents of the transport vesicles mediating endosome to Golgi trafficking. The authors use an in-cellulo vesicle-tethering assay in which golgin vesicle tethering proteins are ectopically localized to mitochondria to capture transport vesicles arriving from endosomes. This is combined with subcellular fractionation and quantitative mass spectrometry to identify those vesicle proteins selectively enriched or shifted towards the mitochondrial fraction, which correspond to constituents of the tethered vesicles. There are differing degrees of enrichment of different vesicle proteins, consistent with different golgins capturing distinct classes of vesicles.

The data are of high quality and overall I found the work to be very convincing. The approach is highly innovative and may be applied to the study of other vesicle trafficking steps. Until now, it has been a major challenge to identify the content of transport vesicles, and the approach shown here indicates that it is feasible, and has identified the complements of endosome to Golgi vesicles with a high degree of confidence. This is an important advance. The study is likely to be of broad interest and I strongly support its publication. There are a few issues that the authors should consider though.

1.) The text mentions 16 proteins rerouted to the GCC88 mitochondria, but, unless I missed it, they are not all named (I could only see six vesicle proteins in figure 4a). As is done for golgin-97, the entire complement of GCC88-tethered vesicle proteins should be listed.

We agree that this was an omission and have now added a new table that gives for all proteins the degree of relocation to mitochondria seen with both Golgin-

97-mito and GCC88-mito, and the statistical confidence (Supplemental Table 2, showing data plotted in Figures 1e and 1f). This table includes a tab listing the 16 hits for GCC88.

2.) The supplementary table is very difficult to follow. It should be reconfigured to make it more legible as to what the different columns and numbers are.

We have improved the labelling of this table so that it is less cryptic, with each cell line and replicate clearly labelled. In addition, to keep the data on a comprehensible scale, the organelle classification and relocation ratios for each protein are now shown in a separate table (Supplementary Table 2).

3.) To exclude the possibility that endosomes are not stuck to the golgin-coated mitochondria, lysosomal hydrolases are studied. The data are consistent with these not being rerouted to the golgin, but it would also be good to mention more typical endosomal markers in this respect e.g. EEA1, Rab5. Presumably these are not shifted to the mitochondrial fractions? If so, it would be good to mention.

EEA1 and Rab5A did not show a detectable shift to mitochondrial fractions, and this is now stated in the text with the values given in Supplementary Table 2.

4.) To make the tomogram (supplementary video) easier to visualize, it would be good to make a 3D reconstruction, showing the volume of the mitochondrion with the vesicles adhered to the surface.

We have now included a 3D reconstruction in the tomogram shown in Supplementary Video 1 and it does indeed make it easier to visualise. We have also included panels from the 3D reconstruction in Figure 4d.

5.) Based upon the Western blot shown in Fig 4h, it is stated that combined KO/KD of all the 3 golgins studied causes reduced cell growth. Cell counting should be done to confirm that growth is indeed reduced.

We agree that cell counting would be better and we have now replaced the Western blot in Figure 4h with a cell counting experiment showing loss of cell growth upon removal of all three golgins.

Reviewer #2 (Remarks to the Author):

The manuscript entitled “Spatial proteomics defines the content of trafficking vesicles captured by golgin tethers” describes an application of LOPIT-DC to identify the content of two different endosome-derived vesicles using a strategy that capture them at the

mitochondria. In this work, the authors used two proteins, golgin-97 and GCC88, to identify cargoes specific to these individual golgins. The paper thus provides an extensive list of cargo and membrane proteins, which includes both known and novel proteins, and support the use of this approach to identify proteins found in different subcellular organelle by tethering them to an ectopic site.

Overall, the manuscript is interesting and demonstrate a novel and elegant application of LOPIT. The data presented is very convincing, and the analysis is very thorough using supervised machine learning and Bayesian statistical modelling. However, the lack of validation of the proteins identified using mutants and by additional control experiments would be necessary to support a physiological relevance of the observed inventory of proteins to make sure that the identification of the protein profiles is not an artefact of targeting organelles to an abnormal destination. The KO presented in figure 4 appear to demonstrate redundancy in those pathways, and do not conclude whether the differences observed by LOPIT when those organelles are tethered to the mitochondria are actually representative of sorting in the golgi.

The threshold used for identifying proteins from the complex consist of the WDR11 complex. While I understand that this complex is a good indicator of proteins found, there could be other proteins with lower enrichment scores that could also be relevant. Considering the analysis presented and the triplicates, I would think that using a proper significant threshold would be more relevant then using known proteins. Are these proteins even considered significantly enriched in their analysis? Is this threshold actually more stringent, or are those proteins simply too close to background?

We agree with the reviewer that proteins with lower enrichment scores could be relevant. The Bayes factor reported determines the relative (marginal) likelihood of the alternative and null. Thus, the larger the Bayes factor the greater the support for the data under the alternative hypothesis. Under standard interpretations of Bayes factors the support of the alternative is **decisive** in our cases (see Knass and Raftery (1995) J Amer. Stat. Assoc. 90, 773-795). Given our Bayesian analysis we refrain from using the word significant as this is a frequentist concept. That being said, significance should be interpreted in terms of the scientific question at hand and is thus why we set biologically defined threshold in our analysis to ensure that the proteins we highlight as being above the threshold are those most worthy of consideration.

To enable other researchers to investigate proteins further down the list we now provide a table which gives the degree of relocation and the Bayes factor for all proteins in the data sets (Supplementary Table S2). In addition, we now provide interactive web apps to visualise the location of a chosen protein in the spatial

map and in the gradient profiles. The links are listed in the Data Availability paragraph, with the app builder Lisa Breckels added as an author.

Other comments:

It is not clear why they used CRAPome to filter out some proteins? It is not explained in the text.

We apologise for not explaining this properly. This filter was applied to remove very abundant cytosolic proteins such as tubulins, keratins and heat shock proteins as they are not organelle specific and can reduce signal-to-noise in LOPIT-based organelle assignments. To clarify this point we have included a tab in Supplementary Table 1 that lists all the proteins removed prior to LOPIT analysis. In addition, the raw fractionation data for these proteins is still included in Supplementary Table 1 so that they can be analysed if others wish to do so.

The supplementary table includes the dataset from the proteomic experiment with the intensity values, but there are no tables with the proteins enriched or grouped from their bioinformatics analysis.

We apologise for this oversight. The complete set of organelle assignments is now shown in Supplementary Table 2 and in the web apps mentioned above, and the sets of proteins specifically associated with the golgin-97-mito and GCC88-mito vesicles are listed in tabs in the same table.

Reviewer #3 (Remarks to the Author):

Previously, the authors showed that re-targeting of golgins to mitochondria is sufficient to also re-distribute Golgi-directed trafficking vesicles (Mie Wong and Sean Munro 2014). Here, in order to unravel the protein content of these vesicles, the authors use innovative methods to isolate mitochondria together with the trapped, mis-localised vesicles followed by MS. Their approach opens interesting new possibilities in the determination of cargo proteins present in small, rare and transient trafficking vesicles that are otherwise difficult to isolate and study. The ultimate aim of these studies is to gain insight in cargo composition of vesicles carrying specific tethering proteins.

Detailed comments:

- As discussed in the text, mito-protein overexpression causes cell stress, which leads to changes in mitochondria morphology. My major worry of the experimental set up is whether the accumulation of some proteins/vesicles nearby the mitochondria could be an artefact due to mito-protein-stress. EM pictures of the negative controls (like overexpression of mito-Tag without golgin protein) would help to understand better the impact of this overexpression. Moreover, silencing of the retromer components should inhibit the recruitment to mitochondria. These controls would improve the sturdiness of the approach.

We apologise for omitting to mention that in our previous studies we have used electron microscopy, proximity biotinylation and direct assays to determine if the mito-Tag results in vesicle accumulation or mitochondrial stress. These studies have found no evidence that the mito-Tag induces mitochondrial stress or causes an accumulation of vesicles (Wong and Munro (2014) Science 346, 1256898; Gillingham et al (2019) eLife 8:e45916). We have now added a comment to this effect.

- The authors claim that there is a general shared result between golgin-97 and golgin-245. It seems to me that this is based only on the observation that both proteins bind AP-1 and TBC1D23. Experiments were done with golgin-97 only. The finding that golgin-97 and golgin-245 have some common interacting partners does not rule out that they could also be on different vesicles. The conclusions on golgin-245 are therefore a bit too speculative.

We agree that it is speculation to assume that everything that applies to golgin-97 also applies to golgin-245, and so we have now removed the references to the later protein wherever the specific data with golgin-97 are discussed.

- The paper is mostly based on the results obtained for golgin-97-mito. There is much less on GCC-88-mito and authors were not able to determine the cargo proteins of these vesicles. Thus, this is a study mostly on golgin-97 rather than a comparison between cargo of vesicles with different tethers. This should be better reflected in the text and title.

The title does not mention GCC88, and the abstract only notes for golgin-97 the number of proteins we identify as being rerouted by the mitochondrial form. We were able to detect fewer proteins that were rerouted by GCC88, but there were some, and in Figures 2 and 4 we show examples of proteins that are rerouted by either golgin-97 alone or by both golgin-97 and GCC88. This seems like useful information about GCC88, and so we would prefer that the abstract still retain a mention of the protein so that the paper can be readily found in PubMed.

- Figure 1a shows a model of the re-targeting system. However, this shows mito-golgin-97-BioID, whereas they refer to a paper where this fusion protein is made without BioID. The model should be made more compatible with the actual experimental set up.

We apologise for this error. Figure 1a has been corrected to show a fusion protein without BioID.

- Figure 1h-i – authors show GCC185-mito staining but this is not cited in the result text.

We now briefly mention GCC185 in the text where Figure 1h-1 is discussed.

- Figure 3g and Figure 4d. Authors claim that cells are selected based on CLEM using fluorescent staining of TMEM87A-RFP and mitotracker. However, I miss here the golgin-97-mito staining. Unless 100% of the cells are co-transfected with TMEM87A-RFP and golgin-97-mito we cannot conclude from the present set of data if golgin-97-mito is indeed present on these mitochondria.

As mentioned above, we know from our previous EM studies that in untransfected cells there is never an accumulation of vesicles around mitochondria. In addition, we show in Figure 4c that, as expected, TMEM87A does not accumulate near mitochondria unless cells are also expressing golgin-97-mito.

- The EM figures show long stretched proteins in between the vesicles. Can the authors comment if these are actually represent the presence of tethers?

This is an interesting idea, and indeed a similar proposal was made about EM images of vesicles found around Golgi stacks (Orsi et al. (1998) PNAS 95, 2279). However, care must be taken in interpreting details of sections of cells that have been chemically stained and embedded in plastic, and so we would rather not speculate about this in the context of this paper.

- PCR projection for LOPIT_DC: the pale coloured bullets are not easy to distinguish.

To make the PCA projections easier to follow we have made the panels in Figure 3 slightly larger and increased the size of the bullets in the legend. In addition, we now provide links to web apps that provide larger and interactive versions of the projections (links provided in the Data Availability paragraph). These have worked well for previous publications on LOPIT from the Lilley lab.

REVIEWERS' COMMENTS

Reviewer #1 (Remarks to the Author):

The authors have done a good job in addressing my comments and I am happy to recommend publication.

Reviewer #2 (Remarks to the Author):

In the revised manuscript from Shin et al., the authors have provided additional experimental data to address the points raised by the reviewers. Overall, the concerns expressed as part of my initial review have been addressed mostly by text revisions and by adding additional data in the tables insomuch that I generally believe the authors have substantially improved the manuscript.

With that said, the authors have not responded to my comment regarding the possibility of adding additional experiments to validate some of the proteins identified. Because of that, there are elements of the original data that remain unclear and do not support the conclusions whether the differences observed by LOPIT when those organelles are tethered to the mitochondria are actually representative of sorting in the golgi.

I believe this is an important point in order to validate both the analyses of their data, but also to support the threshold used. I understand that perhaps it is not reasonable to achieve this in a timely fashion for publication of this article, and the other reviewers have not requested further validation of the experiments.

Reviewer #3 (Remarks to the Author):

The authors have successfully addressed my concerns and I am happy to now recommend this paper for publication

Response to reviewers' comments.

We are very pleased that the reviewers felt that we had done a good job in addressing their comments and concerns, and hence they were happy to recommend publication. To address the remaining concern of Reviewer #2 we have, as requested, added further discussion in the text of potential caveats/artefacts and potential future analysis.

We have also addressed all of the editorial requests, as outlined in our responses in the Author Checklist, and have ensured that the manuscript complies with the policies and formatting requirements of Nature Communications.

Reviewer #2 (Remarks to the Author):

In the revised manuscript from Shin et al., the authors have provided additional experimental data to address the points raised by the reviewers. Overall, the concerns expressed as part of my initial review have been addressed mostly by text revisions and by adding additional data in the tables insomuch that I generally believe the authors have substantially improved the manuscript.

With that said, the authors have not responded to my comment regarding the possibility of adding additional experiments to validate some of the proteins identified. Because of that, there are elements of the original data that remain unclear and do not support the conclusions whether the differences observed by LOPIT when those organelles are tethered to the mitochondria are actually representative of sorting in the golgi.

I believe this is an important point in order to validate both the analyses of their data, but also to support the threshold used. I understand that perhaps it is not reasonable to achieve this in a timely fashion for publication of this article, and the other reviewers have not requested further validation of the experiments.

We have added further discussion in the text to address this issue. The principle aim of our paper was to show that the LOPIT-DC assay works as an approach to characterise the contents of intracellular vesicles, something that we demonstrate by finding that the most highly enriched proteins are known cargo of such vesicles. Although we validate ATG9, TVP23B and furin as being in vesicles captured by golgin-97 but not GCC88, and TMEM87A as being in vesicles in captured by both golgins (and being mislocalised by their absence), these four proteins are the only ones that we validate that had not been previously shown to be relocated by mitochondrial golgins. Thus, to make the caveats about our data clear we have added the following text to the concluding paragraph:

"We have validated four cargo that had not previously been reported as being captured by mitochondrial golgins (ATG9A, TVP23B, furin and TMEM87A). Nonetheless, it should be noted that we applied an empirical cut-off to select a set of high scoring hits for further analysis, and, like any statistical cut-off, it is to some extent arbitrary. Thus, some of the other hits may not be in vesicles, and some of those below the cut-off, especially those close to the boundary, may actually be in vesicles. However, the full list of data provided in the Supplementary tables, and the relative simplicity of the mitochondrial golgin relocation assay, should enable others to readily test the vesicle location of their proteins of interest."